# Volatiles Induced from *Hypolepis punctata* (Dennstaedtiaceae) by Herbivores Attract *Sclomina erinacea* (Hemiptera: Reduviidae): Clear Evidence of Indirect Defense in Fern

**DOI:** 10.3390/insects12110978

**Published:** 2021-10-28

**Authors:** Kerui Huang, Hui Shang, Qiong Zhou, Yun Wang, Hui Shen, Yuehong Yan

**Affiliations:** 1Shanghai Chenshan Plant Science Research Center, Chinese Academy of Sciences, Shanghai Chenshan Botanical Garden, Shanghai 201602, China; huangkerui008@163.com (K.H.); shenhui@sibs.ac.cn (H.S.); yhyan@sibs.ac.cn (Y.Y.); 2College of Life Sciences, Hunan Normal University, Changsha 410081, China; 3College of Life and Environmental Sciences, Hunan University of Arts and Science, Changde 415000, China; wangyun1211@126.com; 4Shenzhen Key Laboratory for Orchid Conservation and Utilization, National Orchid Conservation Center of China and the Orchid Conservation & Research Center of Shenzhen, Shenzhen 518114, China

**Keywords:** electroantennography, GC-MS, *Hypolepis punctata*, indirect defense, olfactory response, *Sclomina erinacea*, volatile organic compounds

## Abstract

**Simple Summary:**

Plants have developed many kinds of indirect defense mechanisms against herbivorous insects, through which the natural enemies are attracted by herbivore-induced plant volatiles to prey on these phytophagous species. There have been many reports regarding this mechanism in spermatophytes, but little is known in ferns. In this study, the relationship between the fern species *Hypolepis punctata* and the predatory insect *Sclomina erinacea* was studied. Employing field observations, plant volatile analysis, electrophysiological experiments, and behavioral experiments, we found that not only the plant can attract the assassin bug in the field, but compared with the healthy individuals, the fern being damaged by herbivorous insects also emitted several novel volatile organic compounds, which were strongly attractive to the assassin bug. The results indicate that ferns may also have indirect defense mechanisms using volatile organic compounds, and it is the first report of an indirect defense mechanism in fern.

**Abstract:**

Plants have evolved various self-defense mechanisms against insect feeding. There are many reports regarding both direct and indirect defense mechanisms in seed-plant. However, only direct defenses on ferns were considered and the indirect defense mechanism has never been reported. In this study, it was observed that the fern *Hypolepis punctata* can attract the assassin bug *Sclomina erinacea* in the field. We collected and analyzed volatiles from *H. punctata* healthy individuals and the ones wounded by *Bertula hadenalis*, using dynamic headspace and GC-MS. We recorded the electroantennogram responses of antennae of *S. erinacea* to different standards of volatile compounds identified from the GC-MS analysis. We also analyzed the behavior of male and female *S. erinacea* adults in response to volatiles collected from *H. punctata* using a Y-tube olfactometer. The results showed that a number of volatile compounds were produced when the fern was damaged by *B. hadenalis*. Electroantennography and Y-tube olfactometer results showed that some herbivore-induced volatiles and volatiles from undamaged leaves could attract *S. erinacea*. Our research suggests that *H. punctata* can attract insect predators by releasing herbivory-induced volatile organic compounds, and for the first time we found ferns may also have indirect defense mechanisms using volatile organic compounds.

## 1. Introduction

The relationship between plants and insects on earth is intricate. Among the approximately one million known insect species, 45% or more are herbivorous [1,2]. To reduce damage from insects, plants have evolved a series of direct or indirect defense mechanisms to prevent insects’ predation [3,4]. For direct defense, physically, plants have thorns on the surface and highly lignified tissues, and chemically, they produce various toxic or repellent metabolites against herbivorous insects [5]. The indirect defense is mainly achieved by attracting beneficial insects (such as predators) from additional trophic levels [6]. Being the most common attractant to these predators, volatile organic compounds (VOCs) secreted by plants are mainly terpenes, fatty acid derivatives, and some other aromatic compounds [7,8,9], they can not only function in the defense process of a single individual, but some can also act as “alarms” to inform other plants to start defense [10,11,12].

Currently, there are many published pieces of research regarding the mechanism of indirect defense in a seed plant. For instance, after being wounded by *Manduca sexta*, *Nicotiana attenuate* can release branched-chain aliphatic acids to attract ants for self-protection [13]. However, for ferns, the focus has been always on direct defense mechanisms. For example, Huang et al. [14] tested extracts from five different parts of ferns and found that they are insecticidal. Markham et al. (2006) showed that proteins from three out of 17 fern species tested have significant toxicity against insects. However, little is known about indirect defense mechanisms in ferns [15].

Using a Gas Chromatography-Mass Spectrometer, Imbiscuso et al. analyzed the composition of volatiles from *Pteris vittata* after it was attacked by *Spodoptera litura* and identified three volatile compounds induced after the feeding, including a terpenoid, suggesting that ferns may also employ indirect defense strategies [16]. However, the research did not delve further into behavioral and electrophysiological experiments using the predators of *S. litura.* Koptur et al. (1998) analyzed relationships between ants and *Polypodium plebeium* in Mexico [17] and found that the damage to the fern by herbivorous insects was significantly reduced only when the fern was producing nectar and ants were present, suggesting that the fern attracts ants (Formicidae) by secreting nectar to protect them from other insects. Although *P. plebeium* attracted ants by secreting nectar, this is unlike plants producing volatile compounds for secondary defense attraction mechanisms, because ants were directly benefited from the nectar. In contrast, the volatiles that attract natural enemies do not provide direct benefit to natural enemies during the indirect defense. Therefore, more studies are needed to elucidate the indirect defense mechanisms in ferns.

Natural enemies play essential roles in plant indirect defense. *Sclomina erinacea*, Hemiptera, Reduviidae is a generalist natural enemy of insects [18] and is mainly distributed in the southeast of China [19]. *Hypolepis*
*punctata* (Dennstaedtiaceae) also grows in the southeast of China, mainly in tropical and subtropical regions, like Hainan, Fujian, Taiwan, Guangdong, Guizhou, Yunnan, Sichuan, Jiangxi, Zhejiang, and Anhui Provinces. It grows in warm, wet environments and can be found in valleys, along the bank of streams, and at the forest edge [20]. We conducted preliminary field studies in Zhangjiajie, Hunan, and Daweishan, Hunan, in 2013, which indicated that *S. erinacea* feeds on the larvae of noctuid moths, including *Callopistria japonica* and *Bertula hadenalis* (Lepidoptera, Noctuidae). In addition, it also hunts for aphids (Aphidoidea), *Drosophila* spp., and other agricultural pests. More interestingly, we found that *S. erinacea* prefers to land on *H. punctata* to hunt Lepidoptera larvae that feed on ferns, such as the larvae of *Callopistria japonica*, and rarely on other ferns. It was also notable that the scent of *H. punctata*, sensed by a human nose, is more discernible than that of many other ferns. Terpenoids are the main attractants to natural enemies in many plants [2,9,21], and in an early study, Hayashi et al. (1977) extracted sesquiterpenes from *H. punctata* [22]. Thus, this characteristic, together with our preliminary field observations, raises the following questions: (1) does *H. punctata* produce attractant volatiles to attract *S. erinacea* to prevent further herbivory from Lepidoptera insects after initial wounding, and (2) if yes, what are the attractive compounds? 

To answer these questions, the volatiles from healthy and wounded *H. punctata* (damaged by larvae of noctuid) were compared using GC-MS and determined the electroantennogram (EAG) responses of antennae of *S. erinacea* to different standards of volatile compounds identified from the GC-MS analysis. The behavior of male and female adults of *S. erinacea* to these volatile compounds was also investigated using a Y-tube olfactometer. Our findings provide insights into the mechanism underlying the attraction of *S. erinacea* to *H. punctata*.

## 2. Materials and Methods

### 2.1. Field Survey of Visitation

A field survey of the visitation and the behavior of *S. erinacea* had been done in an *H. punctata* population, in a valley in southern Hunan province, the *H. punctata* individuals were classified judging by the appearance of the leave into two groups: 41 healthy individuals (HI) and 34 damaged individuals (DI, meaning damaged by phytophagous insects). The *S. erinacea* usually stays on *H. punctata* for hours to a whole day during one visitation, thus the number of *S. erinacea* on each *H. punctata* individual was recorded in the morning. The observation lasted for seven days.

### 2.2. Collecting and Analyzing Volatile Organic Compounds from Hypolepis punctata

VOCs were sampled in naturally growing *Hypolepis punctata* populations in Fengluanxi (110°7′36″ N, 29°19′50″ E), Hunan Province. Several individuals of *H. punctata* with similar growth status and size that showed no sign of insect damage were selected, and the larvae of *Bertula hadenalis*, a species of noctuid were selected as the herbivorous insect. We divided all *H. punctata* individuals into three groups: healthy individuals (HI), undamaged individuals that were growing close to damaged individuals (UICD), and damaged individuals (DI, meaning damaged by *B. hadenalis*). Prior to volatile sampling, one leaf from each individual of *Hypolepis punctata* was safely enclosed in a fine mesh nylon bag. For the DI group, we added two to three two-instar larvae of *Bertula hadenalis* into each bag, and for the HI group, no larvae were placed into the bags. Additionally, the HI group individuals were more than 10 m away from the plants included in the DI group. The individuals of the UICD group were similar to the HI group, as no larvae were added to the bags, but the distance between the UICD individuals and the DI individuals was less than two meters. The mesh bags remained affixed to the leaves for 24 h before volatile sampling. During VOC sampling, the mesh bags and larvae were removed, and each leaf was enclosed in a Nalophan bag (Toppits, Minden, German) with openings at both ends. One end was fastened securely to the fern stipe together with a glass tube (diameter 4 mm, length 15 cm) filled with activated charcoal, and the other end was connected to another glass tube (diameter 4 mm, length 10 cm) containing 30 mg adsorbent (Porapak Q). The end of the adsorbent tube was connected to an airflow meter (model LZB-3W, 100–1000 mL/min, Shanghai Hange Biosciences Corporation, Shanghai), which was connected to an air pump (model QC-1S, Beijing Municipal Institute of Labor Protection, Beijing) set to a flow rate of 400 mL/min for four hours. Each part mentioned was connected using laboratory-grade PVC tubing. In the negative control experiments, the Nalophan bags were empty. In each location, there were four repeats of the experimental groups (DI, HI, and UICD) and two of the negative control group, and tests were conducted concurrently from 9:00 am to 2:00 pm. Compounds were eluted from the adsorbent tube with 600 µL of n-hexane and stored at −20 °C until analyzed.

0.4 µL of the eluent was analyzed using a Shimadzu GC-MS (GCMS-QP2010) with a RESTEK Rtx^®^-5MS column (30 m × 200 mm, film thickness 0.25 mm). The carrier gas used was helium with a flow rate of 2.0 mL/min. The GC-MS was programmed as follows: injector 250 °C, initial column temperature held at 50 °C for 3 min, increased at 10 °C min^−1^ to 210 °C, maintained for 1 min, increased at 20 °C min^−1^ to 250 °C, and held for 3 min. 

We used mass spectra from MS libraries (NIST05, NIST05s, and WILEY7) for tentative peak identification, and only those compounds with a higher concentration than in the control samples were considered. Retention indexes, which were used to convert retention times into system-independent constants, were taken from studies that used the Rtx^®^-5MS column, as well as those that used the similar HP-5MS column. We compared the composition of VOCs in the individuals by calculating proportions of the contribution of identified compounds, which in each individual were calculated by dividing the area of each peak by the total area of all the peaks, except the peaks of contaminants, and multiplying by 100. 

### 2.3. Electroantennogram Recordings

The responses of antennae of both male and female *Sclomina erinacea* to certain volatile compounds of *Hypolepis punctata* were tested using electroantennogram recordings (EAG). Nymphs of *Sclomina erinacea* were collected from Fengluanxi (110°7′36″ N, 29°19′50″ E), Hunan Province, and grown to adults under laboratory conditions fed with the larvae of *Tenebrio molitor* (25 °C, 60% RH). The compounds used for testing (five to be herbivory induced compounds and two green leaf volatiles) were selected according to the results obtained in the VOCs analysis, all the compounds used in this experiment are listed in Table 1. In this experiment, an entire antenna was carefully removed from the head of living adult *Sclomina erinacea*; 4/5 the length of the scape and 1/10 the length of the flagellum was also removed, and both sides of the antenna were attached to electrodes covered with electrode gel (Spectra 360, Parker Labs, Newark, NJ, USA). Ten μL of hexane (as control) or a hexane solution composed of one of the tested seven compounds (10 μg/μL, 100 μg) was dotted on a piece of filter paper (5 mm × 40 mm), which was then inserted into a glass Pasteur pipette. The tip of the pipette was inserted about 4 mm into a small hole in the wall of a glass tube, into which the antenna with the electrodes was inserted 3 cm from the open end of the tube without touching the inner wall. A constant airflow (1 L/min), passing through the glass tube and over the antenna, was generated and controlled by a Syntech CS-55 air stimulus controller, which also controlled the airflow through the pipette. During scent stimulation, an airflow of 40 mL/min was pumped through the Pasteur pipette into the glass tube containing an antenna for 1 s. Each antenna was stimulated two or three times with each compound individually at 60-s intervals, and each compound was repeated using eight different antennae from each sex (male and female). Hexanal was used as the reference standard, which we applied to stimulate the antennae at the beginning and end of each stimulation to correct for the loss of antennal sensitivity. For the correction, we assumed that the decrease of the antennal sensitivity had a linear relationship with time; therefore, data were normalized as follows:(1)$$\mathrm{rEAG}=\frac{\mathrm{EAG}\left(A\right)}{\mathrm{EAG}\left(\mathrm{std}1\right)+\frac{\mathrm{EAG}\left(\mathrm{std}2\right)-\mathrm{EAG}\left(\mathrm{std}1\right)}{\mathrm{RT}\left(\mathrm{std}2\right)-\mathrm{RT}\left(\mathrm{std}1\right)}\times \left(\mathrm{RT}\left(A\right)-\mathrm{RT}\left(\mathrm{std}1\right)\right)}$$
where rEAG stands for relative EAG response; EAG(A) stands for the amplitude (mV) of the EAG response to compound A; EAG(std1) stands for the EAG response to the reference standard at the beginning of each stimulation; EAG(std2) stands for the EAG response to the reference standard at the end of each stimulation; RT(A) stands for the time when the stimulation was done with compound A; RT(std1) stands for the start time of the stimulation using the reference standard at the beginning of each stimulation; RT(std2) stands for the end time of the stimulation using the reference standard.

### 2.4. Y-Olfactometer Experiment

Olfactory behavior of *Sclomina erinacea* towards seven different volatile organic compounds (Table 1) produced by *Hypolepis punctata* was tested using a glass Y-tube olfactometer, and an additional file shows a sample graph of the whole device (Appendix A). Two pear-shaped glass bottles where the odor compounds were placed were connected to each end of the arms of the Y-tube (ID 25 mm; stem 6 cm; arms 15 cm at a 142° angle to the stem). Both the Y-tube and the pear-shaped glass bottles were put inside a top opening square box (length 50 cm; width 35 cm; height 12 cm), made of paper and painted white to reduce environmental disturbance. The contents of the box were illuminated by three 30 W fluorescent lamps. Air cleaned by active charcoal and humidified by passing through distilled water was pumped through each pear-shaped bottle into each arm at a rate of 400 mL/min using an air pump (model QC-1S, Beijing Municipal Institute of Labor Protection, Beijing). Cuboid agar blocks (2%; 1.5 cm length; 1.5 cm width; 0.5 cm height) placed in the glass bottles were used to carry each compound and release them into the Y-tube. Each compound was diluted in hexane. For each test, 5 μL of the solution (10 μg/μL, 50 μg) was injected into one agar block, which was then placed into one of the two pear-shaped glass bottles. Another pear-shaped glass bottle holding an agar block containing 5 μL of hexane was used as the control group. Then, one adult insect was released from the opening of the stem into the Y-tube. Choices were scored when adult insects went 2 cm from the middle of the Y-tube into one arm, and an additional picture shows the position of an adult when it just reached the score point (Appendix A). If an insect did not make any choice after 10 min, then the test was repeated using another insect individual until a choice was finally made. Each compound was tested using at least seven adults of each sex, and each test was repeated three times. To reduce the disturbance of uncertain factors, the Y-tube was cleaned once every five tests using absolute ethanol, and the position of the experimental group and the control group was also exchanged. 

### 2.5. Data Analysis

Data of the field study, VOCs proportions, EAG responses, and the Y-tube tests were analyzed using software SPSS 19.0. G-test was used to analyze the average relative VOCs amounts, Duncan’s multiple range test was used to compare EAG responses triggered by different compounds and G-test was used to compare the selection of different groups in the field study and the Y-tube test.

## 3. Results

### 3.1. The Visitation Observation

The visitation preference difference of *S. erinacea* individuals between the healthy and the damaged *H. punctat**a* in the field observation is shown in Table 2, which demonstrate that, during the seven days of field observations, *S. erinacea* individuals had a strong preference for the damaged *Hypolepis punctata* (*p* < 0.05) by herbivory for five days in the total seven days, indicating that the herbivory of phytophagous insects on *H. punctata* can affect the behavior of the assassin bug *S. erinacea.*

The *p* Values were generated using G-test for the comparison of the Sclomina erinacea individuals’ visitation preference difference between the healthy and the damaged *Hypolepis punctata,* of which the individual numbers were 34 and 41 respectively.

### 3.2. VOCs of Herbivore Damaged and Undamaged Hypolepis punctata

Table 3 presents the relative percentages of each volatile organic compound collected from healthy individuals (HI), undamaged individuals that are close to damaged individuals (UICD), and damaged individuals (DI, meaning damaged by *Bertula hadenalis*) of *Hypolepis punctata*. Twenty-five VOCs were detected in total, with 15 from the HI group, 12 from the UICD group, and 11 from the DI group. The compounds shared among all three groups were α-pinene, 1-octen-3-ol, and nonanal. Six terpenes were detected: α-pinene, β-myrcene, sabinene, α-farnesene, zingiberene, and linalool. Figure 1 illustrates the comparison of relative percentages of nine VOCs from damaged and undamaged *Hypolepis punctata*. The relative percentages (divided by the total area of all the peaks) of five of the nine VOCs released from damaged individuals were greater than those released from healthy individuals, significantly for three of them (β-myrcene, linalool, α-pinene, and nonanal, *p* < 0.05), marginally significantly for α-pinene (*p* = 0.0565), and insignificantly for 1-octen-3-o (*p* > 0.05). Thus, these five compounds (β-myrcene, linalool, α-pinene, nonanal, and 1-octen-3-ol) were assumed to be herbivory-induced volatiles and were selected for the following test in this study.

The other four compounds (hexanal, trans-2-hexenal, 3-hexen-1-ol, and 1-hexanol) are green leaf volatiles (GLVs), and the relative amounts of these four compounds present the form opposite to that of the assumed herbivory induced volatiles. Figure 2 shows the relative percentages of these four GLVs sampled from individuals of the HI, UICD, and DI groups. The individuals of the HI group emitted the most substantial amount of these GLVs, which were released in smaller amounts by individuals of the UICD group, and were the lowest emitted from individuals of the DI group. This indicates that after *H. punctata* were damaged by *B. hadenalis* larvae, the release of some GLVs was reduced.

### 3.3. Electroantennogram Recordings 

Figure 3 illustrates the relative EAG responses of the antennae taken from *Sclomina erinacea* to tested compounds. The antennae of both male and female *S. erinacea* shows the highest responses to trans-2-hexenal compared with other tested chemicals (*p* < 0.05). The elicited relative EAG responses of the male antennae in descending order are trans-2-hexenal, 1-octen-3-ol, nonanal, linalool, hexanal, β-myrcene, α-pinene, and hexane. For antennae removed from female insects, the EAG responses in descending order are trans-2-hexenal, nonanal, hexanal, 1-octen-3-ol, linalool, β-myrcene, hexane, and α-pinene. All chemicals except for α-pinene and β-myrcene elicit significantly higher relative responses in both male and female antennae than hexane (*p* < 0.05). Linalool, α-pinene, trans-2-hexenal, and 1-octen-3-ol elicit higher relative responses in males than in females, and response to 1-octen-3-ol is significantly different (*p* < 0.05). The other three compounds elicit responses that are higher in females than in males (*p* > 0.05). 

### 3.4. Y-Olfactometer Experiment

The number of choices of *Sclomina erinacea* among the seven tested compounds versus hexane using a Y-olfactometer was shown in Table 4, while the percent of choices is shown in Figure 4. To male *S. erinacea*, nonanal, β-myrcene, and trans-2-hexenal showed significant attractiveness (*p* < 0.05, Table 4). As to females, only linalool showed significant attraction (*p* < 0.05, Table 4). However, although the difference was not significant, females prefer trans-2-hexenal (18 vs. 9 times, Table 4), and males prefer linalool (16 vs. 8 times, Table 4) than towards hexane.

## 4. Discussion

Our study provides the first clear evidence of indirect defense mechanisms in ferns. Once wounded by herbivorous insects, *H. punctata* individuals were significantly more attractive to *S. erinacea* compared with healthy individuals in the field (Table 2). Furthermore, the damaged individuals produced a significant amount of novel volatiles compounds, which were not detected in healthy individuals of this fern species. Electroantennogram measurements and behavioral experiments using volatile standards indicated that these novel volatiles compounds attract *S. erinacea.*


### 4.1. Volatiles Induced from H. Punctata

Our results illustrated that there was significantly more β-myrcene, linalool, nonanal, and marginally significantly more α-pinene secreted from wounded *H. punctata* in comparison with intact individuals. The average amount of secreted 1-octene-3-ol was not significantly different between the HI and DI groups; however, the average amount of 1-octene-3-ol was higher from wounded than from healthy plants. Linalool [23], β-myrcene [24], α-pinene [25], nonanal [26], and 1-octene-3-ol [27] have been shown to impact insect behavior. Among them, α-pinene, linalool, and 1-octen-3-ol have been found to be able to attract predators for plant defense [9,28,29]. Therefore, combined the phenomenon with the result of our field observation, we believe that these induced secondary metabolites are volatiles produced in the fern, and they may play an essential role in the indirect defense of *H. punctata* by attracting natural enemies. 

It is metabolically expensive for plants to synthesize induced volatiles, and sometimes the synthesis would result in a reduced or halted production of other common volatiles [21]. In this study, the four green leaf volatiles were detected in only healthy individuals but were not detected in or only detected in small amounts in wounded *H. punctata* or nearby healthy individuals. We speculate that to synthesize induced volatiles, *H. punctata* reduced the release of green leaf volatiles (Figure 4). This is a metabolic trade-off in the plant.

Plants can communicate via volatile chemicals. Studies have shown that when plants are injured, adjacent plants initiate the corresponding mechanism of chemical defense [10,11,12]. In this study, β-myrcene and linalool were found in healthy *H. punctata* individuals near wounded individuals. Wounded H. punctata likely communicated the wounding chemically with other individuals to induce them to produce volatiles. However, this hypothesis needs further exploration. 

### 4.2. Electrophysiological Experiments Using Volatile Standards

Electrophysiological experiments were used to accurately determine whether the insects sense chemicals through smell, which is an excellent method to study the relationship between insects and volatiles. In particular, this allowed us to determine which compounds in the tested volatiles are likely to affect insect behavior [30]. Among the selected volatile standards, five were induced chemicals, and two were GLVs from healthy leaves. All volatiles except α-pinene and β-myrcene generated stronger electrophysiological signals than the control, n-hexane, in the antennae of *S. erinacea*, suggesting that *S. erinacea* may at least sense some of the induced substances produced in the fern and be attracted to assist the fern (the indirect defense mechanism). It is possible that not all volatiles produced by *H. punctata* can be sensed, which might be why two of the identified compounds did not significantly stimulate the antennae. 

Additionally, there might be other unknown functions for these volatiles. Studies have shown that α-pinene is a secondary metabolite with direct defense activity, which has been shown to reduce the number of larvae and adults *Tribolium castaneum* and decrease their food consumption and food utilization rate [31]. To our knowledge, there has been no report on the role of β-myrcene for effective defense in plants. In addition, it is possible that some of the induced components in our study may play a role in the chemical communication between plants of the same species because some of these induced chemical components were detected in individuals near the wounded plants. Green leaf volatiles (GLVs) are a kind of “common smell” emitted by many unwounded plants [32,33]. The releasing of Green leaf volatiles was not insect-wounding-needed. In our study, however, some of the GLVs (e.g., trans-2-hexenal) were able to generate electrophysiological signals in the antennae, and, notably, adult male and female *S. erinacea* were significantly more sensitive to trans-2-hexenal in EAG measurements (*p* < 0.05) than other compounds. This suggests that even before wounding, the insect may be able to sense the presence of *H. punctata* or plants with similar volatiles. This implies that *S. erinacea* may have adapted to sense non-induced plant scents to locate its habitat or proliferation site near potential hunting grounds, and then precisely locate its prey based on the induced chemical compounds, which requires further study.

In addition, there were no significant differences in the EAG measurements of other compounds between male and female *S. erinacea**,* except for with 1-octen-3-ol, suggesting that the induced volatiles may not affect the reproductive behavior of *S. erinacea*, and the leading role of the volatiles is to attract the insects to catch pests.

### 4.3. Behavioral Experiments Using Volatile Standards

Compared with electrophysiological experiments, behavioral experiments can be easily influenced by environmental factors, especially when the tested compounds are unstable; the results are also influenced by the health of the tested insects. Furthermore, it is highly challenging to simulate the natural conditions of the olfactory condition entirely. Even so, among the seven tested volatiles, four had significantly stronger attractive activity on one sex of S. erinacea compared with the control, indicating that the induced volatile compounds attract S. erinacea with more differential activity observed between sexes, especially when compared with the EAG experiments, which is also similarly reported in one other related study [34]. Further investigations are needed to have a better understanding of the behavioral patterns between male and female S. erinacea individuals.

Among the volatile compounds that attract *S. erinacea*, nonanal and 2-hexenal were from undamaged green leaves, and β-myrcene and linalool were induced after wounding. The attractive activity by the remaining three compounds (excluding β-myrcene) is in line with the results of the electrophysiological experiments, suggesting that the attractiveness of *H. punctata* to S. erinacea may be through an indirect defense mechanism. The fact that green leaf volatiles can attract S. erinacea also implies that the insect may establish its habitat based on constitutional (not induced) smells from H. punctata. However, β-myrcene, which had a similar EAG value to hexane, was found to be attractive to adult male *S. erinacea*. This may be due to the reason that fewer odor sensilla that are sensitive to β-myrcene are in the antennae of the adult males than those sensilla sensitive to other compounds. Nonetheless, these sensilla still generate signals that impact the behavior of male *S. erinacea*. More studies are needed to elucidate the specific reasons for the attraction of males to β-myrcene. Our result also suggested that the attractiveness of these volatiles to *S. erinacea* seems to be irrelevant with their relative amounts (Table 4).

Our findings explain our preliminary field observations where we observed that noctuid larvae were heavily preyed on by *S. erinacea* when eating *H. punctata*. However, the effect of this defense mechanism is also dependent on the severity of the pest damage and the population density of *S. erinacea.* Therefore, further studies are needed to investigate the quantitative and dynamic relationship between noctuid larvae and *S. erinacea.*

## 5. Conclusions

Our data provide direct evidence that the herbivory-damaged *H. punctata* has more attractiveness to *S. erinacea* than the healthy individuals in the field, and the volatiles from *H. punctata* can attract *S. erinacea*, especially the induced compounds. This supports the proposition that ferns may also employ indirect defense mechanisms using volatile organic compounds. This mechanism is different from ants being attracted to fern species using nectar as a consumable reward [17] because *H. punctata* does not provide a direct reward to *S. erinacea*, yet still creates a mutually beneficial relationship with *S. erinacea*. Thus, we discovered an indirect defense mechanism of a fern species, and whether the volatile attraction mechanism found in this study resulted from long-term co-evolution still needs further research.

## Figures and Tables

**Figure 1 insects-12-00978-f001:**
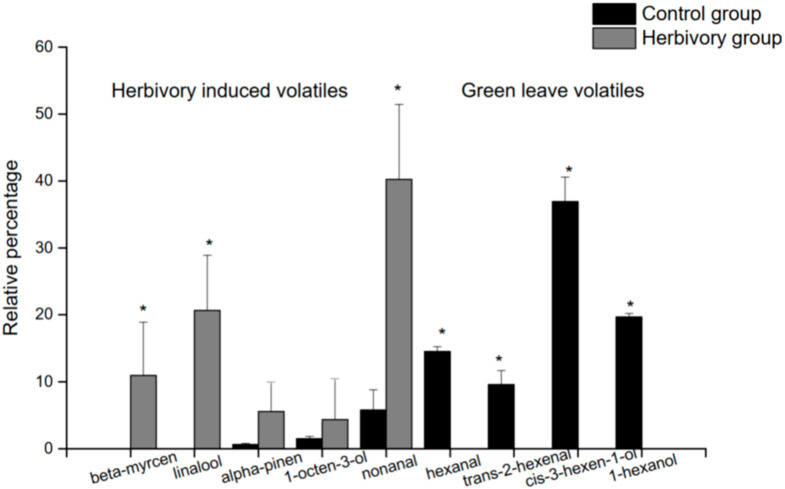
Relative Percentages of Nine VOCs from Damaged and Undamaged *Hypolepis punctata*. The Y-axis is the average relative percentage of the same compound collected from *Hypolepis punctata* individuals of the damaged group and undamaged group; each group is represented with a unique color. The X-axis indicates nine chemical compounds; the five on the left side of the vertical line are assumed herbivory induced compounds, and the four on the right side are green leaf volatiles. * indicates that there is a significant difference between the relative percentage of the same compound collected from the control group (HI group) and herbivory group (DI group) (*p* < 0.05).

**Figure 2 insects-12-00978-f002:**
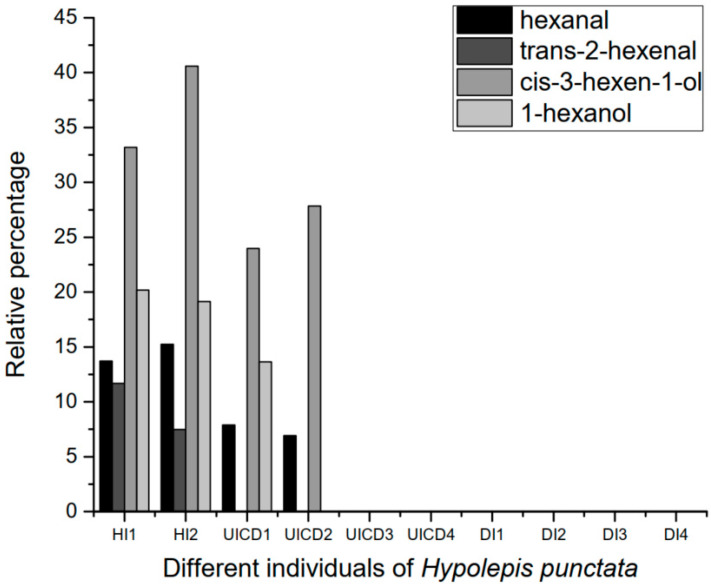
Relative percentages of amounts of four green leaf volatiles in each individual of *Hypolepis punctata*. The Y-axis is the relative percentage of four green leaf volatiles, of which each is represented with a unique color. The X-axis shows ten individuals of *Hypolepis punctata*. HI = healthy individuals, UICD= undamaged individuals that are close to damaged individuals, DI = damaged individuals.

**Figure 3 insects-12-00978-f003:**
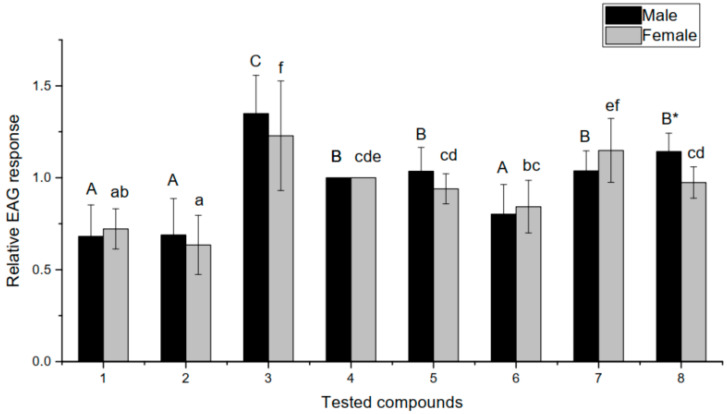
Relative EAG responses of *Sclomina erinacea* antenna to eight compounds. The Y-axis is the average relative EAG response of the antennae of *Sclomina erinacea* to eight selected compounds, which are represented by numbers 1–8 on the X-axis: 1 = hexane, 2 = α-pinene, 3 = trans-2-hexenal, 4 = hexanal, 5 = linalool, 6 = β-myrcene, 7 = nonanal, 8 = 1-octen-3-ol. Each insect sex is represented with a unique color, and the data was analyzed using Duncan’s multiple range test within each sex between compounds. The uppercase or lowercase letters on top of each column are placed for the comparison of the EAG responses of males or females to each chemical, the uppercase letters are for males and the lowercase letters are for females. Columns with one or more common letters on top represent there is no significance (*p* > 0.05) between the EAG responses of antennae of one sex to each corresponding chemicals, otherwise the EAG responses to each corresponding chemicals for each column(s) with unique letter(s) is(are) significantly different from other chemicals. The symbol * means there is significant difference between the EAG responses of males and females to a chemical.

**Figure 4 insects-12-00978-f004:**
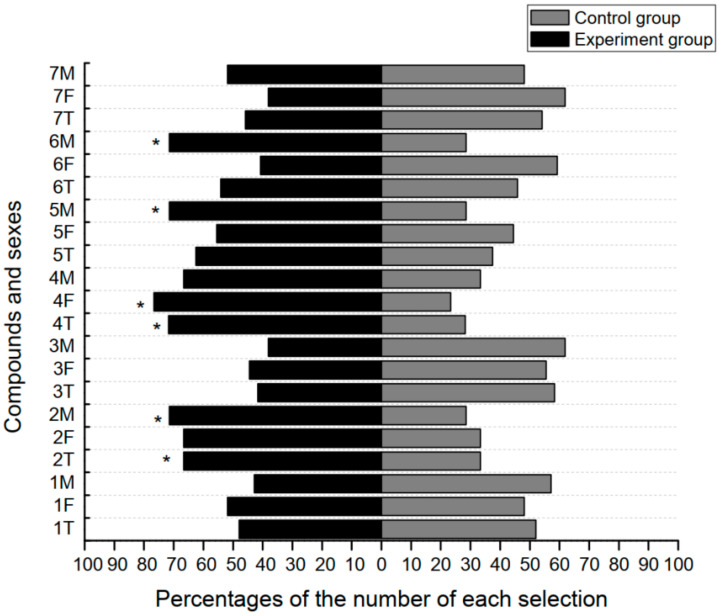
*Sclomina erinacea* individuals’ preference to seven kinds of VOCs produced by *Hypolepis punctata*. Different numbers on the Y-axis represent seven different chemicals: 1 = α-pinene; 2 = trans-2-hexenal; 3 = hexanal; 4 = linalool; 5 = β-myrcene; 6 = nonanal; 7 = 1-octen-3-ol; M = males; F = females; T = total. The proportion of the selection between the control group (hexane group) and the experimental group (different tested chemicals) in each condition on the Y-axis is represented using two different colors. A G-test was used to compare the selection of different groups, and * indicates that there is a significant difference between the choices of the control group and the experimental group.

**Table 1 insects-12-00978-t001:** Standard compounds used for the EAG recordings and Y-olfactometer experiment.

Compounds	Purity	Source
Linalool	97%	AccuStandard^®^
1-Octen-3-ol	98%	AccuStandard^®^
Hexanal	97%	Dr. Ehrenstorfer GmbH
trans-2-Hexen-1-al	98%	Sigma-Aldrich
Nonanal	98%	Dr. Ehrenstorfer GmbH
β-Myrcene	95%	Sigma-Aldrich
α-pinene	98%	Dr. Ehrenstorfer GmbH

**Table 2 insects-12-00978-t002:** The visitation preference difference of *Sclomina erinacea* individuals between the healthy and the damaged *Hypolepis punctata* in the field observation for 7 days.

	Day 1	Day 2	Day 3	Day 4	Day 5	Day 6	Day 7	Total
Damaged individuals	27	30	26	31	19	18	16	167
Healthy individuals	15	11	9	5	6	7	4	57
*p*-Value	>0.05	0.015	0.017	<0.001	0.027	>0.05	0.018	<0.001

**Table 3 insects-12-00978-t003:** Relative percentage of VOCs recorded from the individuals of healthy and *Hypolepis punctata* damaged by herbivory.

Compounds	Relative Percentages of Each Compound in *Hypolepis punctata* Individuals (%)										
	RT	HI1	HI 2	UICD1	UICD 2	UICD 3	UICD 4	DI1	DI 2	DI 3	DI 4
hexanal	3.467	13.73	15.24	7.88	6.93	-	-	-	-	-	-
trans-2-hexenal	4.39	11.67	7.49	-	-	-	-	-	-	-	-
cis-3-hexen-1-ol	4.455	33.19	40.59	23.97	27.86	-	-	-	-	-	-
1-hexanol	4.717	20.19	19.13	13.63	-	-	-	-	-	-	-
alpha-pinene	5.941	0.48	0.81	1.02	1.61	14.41	3.40	5.27	-	14.11	-
2-heptenal	6.494	1.43	-	-	-	-	-	-	-	-	-
benzaldehyde	6.61	1.06	-	-	-	-	-	-	-	-	-
sabinene	6.779	0.38	0.71	-	-	-	-	-	-	-	-
1-octen-3-ol	6.887	1.85	1.15	5.21	6.77	-	9.40	-	-	-	17.93
β-myrcene	7.098	-	-	4.70	5.26	19.28	6.82	8.55	2.88	24.15	8.24
Cis-3-hexen-1-ol, acetate	7.449	1.21	-	-	-	-	-	-	-	-	-
benzeneacetaldehyde	8.171	5.75	-	-	-	-	-	-	-	-	-
sabinene	8.191	-	-	-	-	-	-	4.34	-	-	-
undecane, 3,8-dimethyl-(CAS)	8.321	-	-	3.45	-	-	-	-	-	-	-
acetophenone	8.608	1.66	2.41	-	-	-	-	-	-	-	-
(+)-(R)-*p*-Mentha-1,8-dien-4-ol	8.882	-	-	-	-	-	-	-	-	-	3.50
linalool	9.127	-	-	9.19	7.76	11.39	13.56	12.88	12.78	32.26	24.61
nonanal	9.147	2.78	8.81	20.69	22.24	54.91	66.82	55.91	45.78	29.47	29.72
phenylethyl alcohol	9.466	4.63	-	-	-	-	-	-	-	-	-
delta.-(2)-dodecanol	13.91	-	-	-	-	-	-	13.05	-	-	-
longifolene	14.065	-	3.67	-	-	-	-	-	-	-	-
nerylacetone	14.53	-	-	7.67	15.05	-	-	-	-	-	-
zingiberene	15.07	-	-	2.59	6.53	-	-	-	7.13	-	-
alpha.-farnesene	15.239	-	-	-	-	-	-	-	-	-	16.00
phenol, 3,5-bis (1,1-dimethylethyl)-	15.341	-	-	-	-	-	-	-	31.43	-	-

RT = retention time, HI = healthy individuals, UICD = undamaged individuals close to damaged individuals, DI = damaged individuals. Sample HI3 and HI4 were damaged during transportation, so data is missing.

**Table 4 insects-12-00978-t004:** *Sclomina erinacea* individuals’ preference in Y-olfactometer experiment to seven kinds of VOCs produced by *Hypolepis punctata* versus hexane.

Compounds	Average Relative Percent in DI Group (%)	Sex	Number of Choices to Experiment Group	Number of Choices to Control (Hexane) Group	*p*-Value
linalool	20.63	Total	39	15	* 0.0013
		Female	23	7	* 0.0027
		Male	16	8	0.0992
β-myrcene	10.96	Total	30	18	0.0816
		Female	15	12	0.5633
		Male	15	6	* 0.0459
alpha-pinene	4.85	Total	23	25	0.7728
		Female	14	13	0.8474
		Male	9	12	0.5120
hexanal	0.00	Total	20	28	0.2471
		Female	12	15	0.5633
		Male	8	13	0.2729
1-octen-3-ol	4.48	Total	22	26	0.5635
		Female	8	13	0.2729
		Male	14	13	0.8474
trans-2-hexenal	0.00	Total	33	15	* 0.0085
		Female	18	9	0.0803
		Male	15	6	* 0.0459
nonanal	40.42	Total	26	22	0.5635
		Female	11	16	0.3345
		Male	15	6	* 0.0459

A G-test was used to compare the selection of different groups, and * indicates that there is a significant difference (*p* < 0.05) between the choices of the control group and the experimental group.

## Data Availability

The datasets used during the current study are available from the corresponding author upon request.

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
