# Peer review of "Volatiles Induced from Hypolepis punctata (Dennstaedtiaceae) by Herbivores Attract Sclomina erinacea (Hemiptera: Reduviidae): Clear Evidence of Indirect Defense in Fern"

_insects, 2021, doi:10.3390/insects12110978_

Round 1

Reviewer 1 Report

Review ID 1413883

Volatiles induced from Hypolepis punctata (Dennstaedtiaceae) by herbivores attract Sclomina erinacea (Hemiptera: Reduviidae): clear evidence of indirect defense in fern

               Volatile compounds represent the future in plant protection. Their role is still underestimated. Combined with other defense mechanisms, they can become the main pest control strategy. Not only direct defense but also indirect defense is of great importance.

               This is quite well organized manuscript. I found this “ms” interesting and innovative. However, a few questions must be explained more precisely.

Critical review:

  1. Title:

It is very clear and can stay in present form.

  1. Introduction:

If the article concerns volatile organic compounds, please write a paragraph regarding these components. Literature is huge. It is worth presenting a thought regarding their role in the future in plant protection. Why do we actually study the relationship between plants and insects based on volatile compounds?

  1. I do not see a clearly defined research aim of the study.

4.     Materials and Methods.You wrote:We divided all individuals into three groups: healthy individuals (HI), undamaged individuals that were growing close to damaged individuals (UICD), and damaged individuals (DI, meaning damaged by Bertula hadenalis).

Exposing undamaged plants in the vicinity of damaged plants is called “plant priming”. Where is information about this phenomenon in the manuscript?

  1. Why are you testing nonanal? What is the sense?
  2. The Y-tube olfactometer is a well-known tool for working with insects and the selection test. However, other olfactometers exist. Why didn't you use a 4-arm olfactometer for your experiments?
  3. Results and Disscussion are well presented.

8.     Conclusions are unacceptable. Of course, they show everything that has been studied, but why another long description? Conclusions must be short. I propose to present 3 points that will reflect the whole meaning of the article. 

Some other papers to add:

  • Orientation of European corn borer first instar larvae to synthetic green leaf volatiles. Journal of Applied Entomology 137(3): 234-240; 2013.

DOI: 10.1111/J.1439-0418.2012.01719.X

  • Volatile organic compounds released by maize following herbivory or insect extract application and communication between plants. Journal of Applied Entomology 141: 630–643; 2017.

DOI: 10.1111/JEN.12367

  • Sitophilus granarius responses to blends of five groups of cereal kernels and one group of plant volatiles. Journal of Stored Products Research 62: 36-39; 2015.

DOI: 10.1016/J.JSPR.2015.03.007

Reviewer 2 Report

The study: Volatiles induced from Hypolepis punctata (Dennstaedtiaceae) 
by herbivores attract Sclomina erinacea (Hemiptera: Reduviidae): clear evidence of indirect defense in fern
 by Kerui Huang , Hui Shang , Qiong Zhou , Yun Wang , Hui Shen , Yuehong Yan, its very interesting, and the central point of the study is clear, besides presenting the indirects mecanisms  of defense, employing field obversation, plant volatile analysis, electrophysiological experiments and behavioral experiments. Nonetheless, despite the authors have tried to clarify the metodology used along the text, This one lacks illustrations that help the understanding of what was done and can be attached in supplementary material.

In the methodology the plants were colected in only one place, the valley. What is the extension of the valley ? Did the species distribuited along the local? Clarify!

line 319, the authors reported an field observation, preliminary, related to the Spodoptera larvae, that is out of context. I recommend the authors to rewrite with cientifical basis. By the end of this paragraph, the authors comment: Therefore, further studies are needed to investigate the quantitative and dynamic relationship between Spodoptera larvae and S. erinacea." but that is not the object of this study.

Reviewer 3 Report

The authors collected the odours from ferns either healthy or damaged by insects in the field and experimentally and found that some plant volatiles were attractive to an insect predator.

The question addressed about the indirect defence mechanism and experiments performed by the authors are original and as mentioned never been done in ferns. However the approach that took the authors is a bit confusing. They first assessed the presence of predators preferentially on damaged plants, without controlling the type/origin of predation, and then organized a manipulated experiment in the lab with larvae of a moth and found some differential compounds between damaged and healthy plants as well as electrophysiological and behavioural responses differences between the volatile extracts or corresponding synthetic compounds. But why authors did not test directly with their behavioural set up (y-olfactometer) or in the field, the attraction of the predatory insects to plants directly damaged by these moth larvae? Authors could use a bigger volatile chamber to set-up a healthy plant vs. a plant with larvae connected to their y tube? Or set up small insect cages on ferns in the field and measure visit from the predatory insects. A direct choice assays between damaged and undamaged ferns would have been better. So yes I agree that their results “provide insights” only (L.104).

Also, authors mentioned for the first time in the discussion their preliminary findings with another moth (spodoptera), why did not they use this moth instead? Lack of lab colony?

The introduction is comprehensive enough, providing background and justification, explaining the gaps in literature about the indirect defence mechanisms in ferns.

It was interesting to see that authors included a treatment category for the undamaged individuals growing close to damaged individuals (UICD). This is unusual and interesting to look out for plant-plant communication, that authors discussed briefly at the end.

Also, I appreciated that authors provided field observations and lab experiments to test their hypothesis.

Authors discussed well there results and suggested several new avenues of research to further explore and support their findings.

Methods:

L .142: did authors concentrated their extracts before GCMS analysis?

Authors calculated proportions of each peak within each sample, so each coumpound is correlated within the sample, requiring multivariate stat analysis.

  1. 161: the predators were maintained with larvae of Tenebrio in the lab, do they think this could have influenced future choice preference?

L.220 : you compared VOCs relative proportions not amount, so you should use other test not ANOVAs. Unless you provide some support for it.

Results:

Table 3: why are they only 2 healthy individuals reported? (in methods L.138, author mentioned that there were 4 repeats of each experimental groups). Please provide explanation why 2 HI were omitted in the text of Data analysis (not only underneath the table).

Figure 1. is control group = the empty bag or healthy plants or HI & IUCD? Please precise and use same terminology across the text.

L- 251-252: careful interpretation in results, move to discussion.

Figure 4: I am confused with terminology of control group and experimental group, do authors mean control compound vs. tested compounds? (I would not use the term group here but instead compound). Also, I would remove the total comparison as long as you present both females and males. There is no added value to combined the two, to my opinion.

Discussion

L.319-323 does not relate to your actual study, so this should not be place in your first paragraph of your discussion, but later. Otherwise we wonder why the authors did not choose larvae of Spodoptera for their experimental treatments.

Table 4 is a duplicate of Figure 4, so I would keep only one. I would suggest to keep the table as it looks clearer than the figure.

Also, please write all your text with the past tense and not a mix of two tenses.

Some minor comments among others written as comments in the pdf:

  1. 21: species

L.26: results

  1. 31: considered

L.58 reformulate to : “many research published on..”

L449: their

Add dates to references in the text. (if you use endnote, you can use “/” to have the author in the text and the date in brackets).

Author Response

This manuscript is a resubmission of an earlier submission. The following is a list of the peer review reports and author responses from that submission.